# Neutrophil-Percentage-to-Albumin Ratio as a Predictor of Coronary Artery Ectasia: A Comparative Analysis with Inflammatory Biomarkers

**DOI:** 10.3390/diagnostics15131638

**Published:** 2025-06-27

**Authors:** Mehdi Karasu, Şeyda Şahin

**Affiliations:** Department of Cardiology, Fethi Sekin Sehir Hastanesi, Elazıg 23280, Turkey; seydshn.58@gmail.com

**Keywords:** coronary artery ectasia (CAE), neutrophil-percentage-to-albumin ratio (NPAR), inflammation, highly sensitive C-reactive protein (hsCRP), neutrophil-to-lymphocyte ratio (NLR), platelet-to-lymphocyte ratio (PLR), Markis classification, systemic inflammation

## Abstract

**Background/Objectives**: Coronary artery ectasia (CAE) is characterized by abnormal dilation of the coronary arteries and is associated with adverse cardiovascular events. Inflammation is believed to play a pivotal role in the development and progression of CAE. The neutrophil-percentage-to-albumin ratio (NPAR) has emerged as a novel marker of systemic inflammation and may serve as a useful tool in the evaluation of CAE. This study aimed to assess the association between the NPAR and CAE and compare its predictive value to established inflammatory biomarkers, including highly sensitive C-reactive protein (hsCRP), the neutrophil-to-lymphocyte ratio (NLR), and the platelet-to-lymphocyte ratio (PLR). **Methods**: A retrospective analysis was conducted on 5212 patients who underwent elective coronary angiography between March 2019 and March 2023. The cohort included 165 patients with isolated CAE and 180 controls with normal coronary anatomy. Inflammatory markers and their correlation with CAE were analyzed using logistic regression models and receiver operating characteristic (ROC) analysis to determine predictive performance. **Results**: The NPAR was significantly elevated in the CAE group compared to the controls (*p* < 0.001). Multivariate analysis identified the NPAR (OR: 2.14, *p* = 0.003) and CRP (OR: 1.53, *p* = 0.02) as independent predictors of CAE. ROC analysis demonstrated that the NPAR had superior predictive value over CRP (AUC: 0.725 vs. 0.635). Additionally, the NPAR showed a strong correlation with CAE severity based on the Markis classification, with higher NPAR values associated with more advanced disease. **Conclusions**: The NPAR is an independent predictor of CAE and outperforms CRP in predicting both the presence and severity of the condition. As a cost-effective and accessible biomarker, the NPAR may enhance the clinical assessment of CAE and provide valuable insights into its inflammatory underpinnings. Further prospective studies are warranted to validate these findings and explore the potential of the NPAR in risk stratification and management of CAE patients.

## 1. Introduction

Coronary artery ectasia (CAE) is defined by an abnormal dilation of the coronary arteries, characterized by either localized or diffuse expansion, where the luminal diameter is at least 1.5 times larger than the adjacent normal segment [1]. This abnormal dilation can lead to clinical complications, such as angina pectoris and myocardial infarction, even in the absence of underlying coronary artery disease (CAD), as a result of vasospasm, dissection, or thrombus formation [2]. Understanding the factors contributing to the presence and severity of CAE is crucial for improving the management of affected patients. Markis et al. [3] suggested that the degradation of the tunica media, often accompanied by inflammatory cell infiltration in ectatic segments, plays a key role in the pathogenesis of CAE [4].

Various studies have explored the involvement of inflammatory biomarkers in CAE, including C-reactive protein (CRP) [5], the platelet-to-lymphocyte ratio (PLR) [6], and the neutrophil-to-lymphocyte ratio (NLR) [7]. Recently, the neutrophil-percentage-to-albumin ratio (NPAR) has emerged as a promising marker of both systemic and localized inflammation [8]. This composite measure, which combines neutrophil percentage with albumin levels, may offer a more nuanced evaluation of the inflammatory response.

The NPAR has gained increasing recognition for its prognostic relevance in several clinical conditions [9,10,11]. A study conducted in China revealed that the NPAR is associated with all-cause mortality in chronic heart failure patients and outperforms albumin or neutrophil percentage alone in predicting outcomes [12]. Similarly, Sun et al. demonstrated that the NPAR is an independent predictor of all-cause mortality in critically ill CAD patients, showing greater predictive strength than its individual components [13]. An elevated NPAR has also been linked to higher mortality rates in atrial fibrillation [14], acute myocardial infarction [15], and cardiogenic shock [16]. Despite these encouraging findings, there is still a significant gap in the literature: no studies have directly compared NPAR with established inflammatory markers in the context of CAE.

The association between the NPAR and CAE may reflect underlying inflammatory and immune dysregulation. Neutrophil predominance indicates heightened innate immune activation, contributing to endothelial damage and vascular remodeling. Concurrent hypoalbuminemia, a negative acute-phase reactant, reflects systemic inflammation and poor vascular integrity, potentially facilitating coronary dilation.

The goal of our study is to investigate the relationship between the NPAR and CAE and to compare its predictive value with other inflammatory markers, such as the NLR, the PLR, and CRP. By evaluating the NPAR within this framework, we aim to identify a novel approach for assessing CAE and broaden our understanding of the role of inflammatory markers in this condition.

## 2. Methods

### 2.1. Patients and Methods

#### 2.1.1. Study Design and Population

This study received approval from the Fırat University Ethics Committee (Ethics Committee acceptance number: 16817), adhering to the International Code of Ethics and the Declaration of Helsinki. In this retrospective case–control study, we screened a total of 5212 patients who underwent elective coronary angiography between March 2019 and March 2023. Patients who underwent percutaneous coronary intervention or coronary artery bypass grafting prior to this study were excluded. Additional exclusion criteria included left ventricular systolic dysfunction (LVEF < 40%), evidence of acute or chronic infections, systemic inflammatory or autoimmune diseases, a history of liver disease, and clinically significant endocrine, hematologic, respiratory, or metabolic disorders and malignancies. After the exclusion criteria, 165 patients diagnosed with isolated CAE were identified.

The control group comprised 180 patients with angiographically normal coronary arteries, matched for age and sex. Baseline demographic characteristics and relevant clinical information were extracted from the hospital’s electronic medical records. In total, our retrospective analysis involved a cohort of 345 eligible patients. A flowchart detailing our study design is illustrated in Figure 1.

We performed a detailed review of the patients’ medical records and did not find significant differences in the use of medications that could influence inflammatory markers, such as corticosteroids, statins, or anti-inflammatory drugs, between the CAE group and the control group.

#### 2.1.2. Data Collection

Data were collected through a review of electronic medical records, which included admission notes, laboratory results, angiographic images, and clinical evaluations. Arterial hypertension was defined by repeated blood pressure readings exceeding 140/90 mm Hg or the current use of antihypertensive medications. Diabetes mellitus was diagnosed based on fasting plasma glucose levels of ≥126 mg/dL on multiple occasions or the use of anti-diabetic medications. Data collection was conducted in accordance with ethical and privacy guidelines, ensuring patient confidentiality and anonymity.

#### 2.1.3. Angiography Evaluation

Coronary angiography was performed via the femoral or radial artery using the Judkins technique. All angiograms were recorded in DICOM digital format at a rate of 25 frames per second. The angiographic images from the eligible patients were evaluated by two experienced interventional cardiologists who were blinded to the patients’ clinical conditions to minimize bias. In instances of disagreement during the visual assessments, a third independent observer was consulted to reach a consensus.

#### 2.1.4. CAE Assessment

CAE was defined as dilation of the coronary arteries, either localized or diffuse, with a luminal diameter at least 1.5 times larger than that of the adjacent normal segment [1]. Isolated CAE was characterized by the absence of coronary artery stenosis, and the severity of isolated CAE was classified according to the Markis classification [3]. In decreasing order of severity, type 1 was defined as diffuse ectasia in two or three vessels, type 2 as diffuse disease in one vessel and localized disease in another, type 3 as diffuse ectasia in only one vessel, and type 4 as localized segmental ectasia.

#### 2.1.5. Laboratory Measurements

Peripheral venous blood samples were collected from the patients upon their admission to the inpatient ward. An automated blood cell counter (Beckman Coulter analyzer, Brea, CA, USA) was employed to measure complete blood count parameters. Blood biochemistry parameters, including creatinine, total cholesterol, high-density lipoprotein (HDL), low-density lipoprotein (LDL), and total bilirubin levels, were measured. hs-CRP levels were assessed prior to coronary angiography using the nephelometric method with an automated analyzer (Beckman Coulter analyzer). The NPAR was calculated by dividing the neutrophil percentage by the albumin level, while the NLR and PLR were determined by dividing the neutrophil count by the lymphocyte count and the platelet count by the lymphocyte count, respectively, using the same blood sample.

#### 2.1.6. Left Ventricular Ejection Fraction (LVEF) Measurement

LVEF was calculated using the modified Simpson’s method based on end-diastolic and end-systolic apical two- and four-chamber views.

#### 2.1.7. Statistical Analysis

Statistical analyses were conducted using SPSS version 22.0 (IBM Corp, Armonk, NY, USA). The normality of continuous variables was assessed using the Kolmogorov–Smirnov test. Normally distributed variables are expressed as mean ± SD, and comparisons were made using the independent t-test. Non-normally distributed variables are presented as median (IQR), and comparisons were made using the Mann–Whitney U test. For comparisons involving more than two groups, ANOVA was utilized. In instances of significant differences across multiple classes of Markis classification, post hoc multiple comparisons were conducted, applying the Bonferroni correction to mitigate the increased risk of Type I error from multiple testing. Non-normally distributed data are reported as medians (interquartile range) and were analyzed using the Mann–Whitney U test. Binary variables are presented as percentages and were analyzed with the Chi-square test. Pearson’s correlation coefficient was utilized to examine the relationship between the NPAR and CAE. Univariate and multivariate logistic regression analyses were conducted to identify independent predictors of CAE, allowing for adjustments for potential confounding variables and determining the unique contribution of each predictor to the likelihood of CAE. All variables with a *p*-value < 0.05 in the univariate logistic regression were considered candidates for the multivariate model. A stepwise backward elimination method was employed to identify independent predictors. Variables with known clinical relevance were retained in the model regardless of statistical significance to ensure robustness. The predictive performance of the NPAR was evaluated using receiver operating characteristic (ROC) curve analysis and the area under the curve (AUC). This analysis provided insights into the diagnostic ability of biomarkers, assessing their sensitivity and specificity across different threshold levels. The optimal cut-off value was identified using Youden’s index, which maximized the difference between true positive and false positive rates, effectively distinguishing between patients with and without CAE. A *p*-value of <0.05 was considered statistically significant.

## 3. Results

The study population had a mean age of 60 ± 10.5 years, with 235 patients (68.1%) being male. Table 1 summarizes the demographic and laboratory data of all the participants.

There were no significant differences in the prevalence of comorbidities such as hypertension, diabetes, and smoking status between the groups. Both groups exhibited similar white blood cell (WBC) and hemoglobin levels. However, the levels of neutrophils, CRP, the NPAR, the NLR, and the PLR were significantly elevated in the ectasia group (*p* < 0.001), while albumin and lymphocyte levels were notably higher in the group with normal coronary anatomy (*p* < 0.001). The right coronary artery (RCA) was the most frequently affected vessel, followed by the left anterior descending artery (LAD) and the circumflex artery (Cx).

In the univariate logistic regression analysis, the neutrophil count, lymphocyte count, albumin, CRP, NPAR, NLR, and PLR levels showed significant associations with isolated CAE. In the subsequent multivariate logistic regression analysis, the NPAR and CRP were identified as independent and significant predictors of isolated CAE (Table 2).

The receiver operating characteristic analysis comparing the predictive values of CRP and the NPAR (Figure 2) revealed that the NPAR had a superior positive predictive value compared to CRP.

Using DeLong’s test, the AUC for the NPAR (0.725) was significantly greater than that for hsCRP (0.635, *p* = 0.03), confirming the superior discriminatory ability of the NPAR in predicting CAE. An NPAR threshold of 1.45 was found to predict isolated coronary ectasia with a sensitivity of 76% and a specificity of 60%.

When evaluating the variables that showed significant differences across the Markis classification, notable variations were observed in the NPAR, NLR, PLR, lymphocyte, and neutrophil levels among the different Markis classes (Table 3).

After conducting multiple comparisons with the Bonferroni correction, we noted the following:A statistically significant difference was identified between the Markis 1 and Markis 4 groups for the NPAR, with a *p*-value of 0.002.For neutrophils, significant differences were noted between Markis 1 and Markis 4, as well as between Markis 3 and Markis 4, with *p*-values of 0.006 and 0.046, respectively.Regarding lymphocytes, significant differences were found between Markis 1 and Markis 4 and between Markis 2 and Markis 4, with *p*-values of 0.002 and 0.032, respectively.For the NLR, a significant difference was observed between Markis 1 and Markis 4, with a *p*-value of 0.001.Significant differences in the PLR were identified between Markis 1 and Markis 4, as well as between Markis 2 and Markis 4, with *p*-values of 0.003 and 0.025, respectively.

As illustrated in Figure 3, patients in lower Markis classes exhibited significantly higher NPAR values compared to those in higher classes.

The correlation analysis between the NPAR and Markis grades indicated a significant inverse relationship, with the NPAR correlating significantly with Markis grade (*p* < 0.001, r = −0.301).

## 4. Discussion

This study explored the association between the NPAR and CAE, demonstrating that the NPAR, along with NLR, PLR, neutrophil, and CRP levels, was significantly elevated in CAE patients compared to those with normal coronary anatomy. These findings reinforce the notion that inflammatory processes, indicated by an elevated NPAR, may play a pivotal role in the pathogenesis of CAE. Our results align with previous studies that have implicated inflammation as a key driver in the development of CAE.

The independent predictive value of the NPAR for CAE, as demonstrated in our study, supports the growing body of evidence linking inflammatory biomarkers to CAE. Previous research has identified the NLR, the PLR, and hs-CRP as significant inflammatory markers associated with CAE. Cannata et al. recently demonstrated that an elevated NLR is an independent predictor of poor outcomes in patients with acute myocarditis, even among those with preserved ejection fraction—traditionally considered a lower-risk group. Their multicenter analysis reinforces the utility of the NLR as a powerful inflammatory biomarker capable of stratifying cardiovascular risk across a broad spectrum of conditions, including those driven by immune dysregulation [17]. For instance, a meta-analysis [18] involving 1775 CAE patients and 1,485 controls revealed significantly higher NLR levels in CAE patients (SMD = 0.73; 95% CI: 0.27–1.20, I^2^ = 97%). Similarly, this analysis found significantly elevated hs-CRP levels in CAE patients (SMD = 0.96; 95% CI: 0.64–1.28, I^2^ = 94%).

While some studies, like those of Çagırcı G. et al. concluded that biomarkers such as TNF-α, IL-6, hs-CRP, and NLR may not independently predict CAE [19], a conclusion also drawn by Çekici Y. et al. [20], others, including Demir M. et al. [21] and Yılmaz M. et al. [22], supported NLR as an independent predictor of CAE. More recently, Fan CH. et al. [23] found that only IL-6 and hs-CRP were independent predictors of CAE, while Tosu AR et al. reported no independent predictive role for the NLR in their cohort [24]. In contrast, Kundi et al. proposed the PLR as an independent predictor of CAE [6].

Our study adds to this literature by showing that while the NLR, the PLR, CRP, and the NPAR are all elevated in CAE patients, only CRP and the NPAR appear to be independent predictors of the condition. Notably, the NPAR outperformed hs-CRP as a predictive marker, as evidenced by a higher area under the curve (AUC: 0.725 vs. 0.635) in our ROC analysis.

Additionally, the observed association between the NPAR and the Markis classification further supports the link between inflammation and disease progression. Previous studies have shown mixed results regarding the role of inflammatory markers in assessing CAE severity. Kundi et al. [6] found an association between PLR and the severity of isolated CAE, while Shereef AS. et al. [25] linked hs-CRP, the PLR, and the NLR to Markis classification, reporting hs-CRP as the only independent predictor of CAE. Similarly, Yalçın AA. et al. [26] identified a positive correlation between the NLR and CAE severity, while Liu R. et al. [27] reported no such association for the NLR or hs-CRP. Conversely, Sarli B. et al. [7] concluded that both hs-CRP and the NLR are independent predictors of CAE severity. Our findings extend these insights to the NPAR, suggesting that it may not only predict the presence of CAE but also correlate with its severity. The significant difference in the NPAR between the mildest (Markis 4) and most severe (Markis 1) forms of CAE highlights its potential for risk stratification in clinical settings.

The elevation in neutrophil counts and inflammatory ratios (NLR, NPAR) in our CAE cohort aligns with previous research indicating a role for neutrophil-mediated inflammation in CAE. Neutrophils are integral to both acute and chronic inflammatory processes, and their activation has been implicated in vascular wall injury, potentially contributing to coronary artery dilation. The differences in neutrophil levels across Markis classifications in our study are consistent with earlier reports linking higher neutrophil counts to more extensive CAE [26].

Interestingly, we found lower albumin levels in the CAE group compared to the individuals with normal coronary anatomy. Hypoalbuminemia is often associated with chronic inflammation and poor nutritional status, both of which are risk factors for vascular abnormalities [28]. Low albumin levels can contribute to complications, such as endothelial dysfunction and platelet aggregation, potentially leading to coronary artery stenosis [29,30]. This inverse relationship between albumin and CAE was previously reported by Sercelik et al., who noted a negative correlation between serum albumin levels and CAE [31]. Our findings confirm that reduced albumin, when combined with elevated neutrophils, likely amplifies the inflammatory environment in CAE patients.

Moreover, we observed decreased lymphocyte levels in patients with isolated CAE, a finding that could be attributed to increased stress-related steroid production and enhanced lymphocyte apoptosis due to elevated inflammation [32,33]. Reduced lymphocyte counts have also been reported in acute cardiovascular events, with studies noting a negative correlation between lymphocyte levels and cardiovascular prognosis [34].

From a clinical standpoint, the NPAR presents several advantages over traditional biomarkers, such as CRP, the NLR, and the PLR. It offers a more comprehensive assessment of inflammation by combining two key components: neutrophils and albumin. Furthermore, it is a cost-effective and accessible biomarker. Our analysis showed that the NPAR could predict CAE with 76% sensitivity and 60% specificity at a cut-off value of 1.45, making it a valuable tool for identifying patients at risk of CAE in everyday clinical practice.

Indeed, the significant association between the NPAR and both the presence and severity of CAE highlights its potential utility in clinical practice. By integrating NPAR assessment into routine practice, clinicians may improve patient outcomes through more personalized management strategies that target inflammation in CAE. Additionally, investigating the therapeutic potential of targeting these inflammatory pathways could unveil new avenues for CAE treatment.

Regarding the use of anti-inflammatory medications in patients with elevated NPAR values, we agree that this is an intriguing area for future research. While our study did not directly evaluate the impact of anti-inflammatory therapies on NPAR or CAE outcomes, the established role of inflammation in CAE pathogenesis suggests that targeting inflammation could be beneficial. However, we emphasize that such interventions should be guided by randomized controlled trials to establish their efficacy and safety in this specific patient population. Markers such as endothelin-1, circulating endothelial cells, or vascular cell adhesion molecules may provide more specific insights into the pathophysiology of CAE and should be evaluated in future prospective studies.

Despite these encouraging results, our study has limitations. As a cross-sectional analysis, it cannot establish a causal relationship between the NPAR and CAE. Additionally, our study population was predominantly male, which may limit the generalizability of the findings. The retrospective design may introduce selection bias, particularly given the single-center nature of this study and the use of angiography-based recruitment. Future prospective studies involving larger and more diverse populations are required to validate the role of the NPAR in diagnosing and stratifying CAE risk.

## 5. Conclusions

In this study, we demonstrated that the NPAR is significantly associated with both the presence and severity of CAE, surpassing hs-CRP as a predictive biomarker. The strong correlation between the NPAR and the Markis classification underscores its potential role in assessing CAE progression. These findings contribute to the expanding literature on the role of inflammation in CAE and suggest that the NPAR could be a valuable addition to the current repertoire of biomarkers used to evaluate patients with suspected coronary abnormalities.

By elucidating the underlying inflammatory mechanisms associated with CAE and validating the NPAR as a superior biomarker, we can enhance the clinical management of this condition and potentially improve patient outcomes. Given the increasing recognition of inflammation’s role in cardiovascular disease, the NPAR presents a promising avenue for future research and clinical application in the context of coronary artery abnormalities. While the NPAR shows promise as a diagnostic and prognostic biomarker for CAE, prospective multicenter studies are necessary to validate its clinical utility before routine application.

## Figures and Tables

**Figure 1 diagnostics-15-01638-f001:**
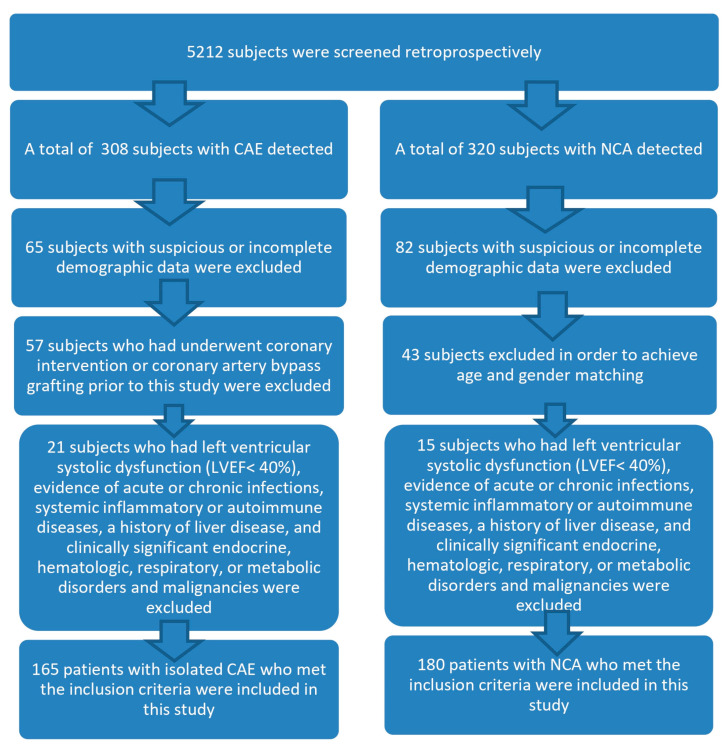
Flow-chart diagram of subject inclusion.

**Figure 2 diagnostics-15-01638-f002:**
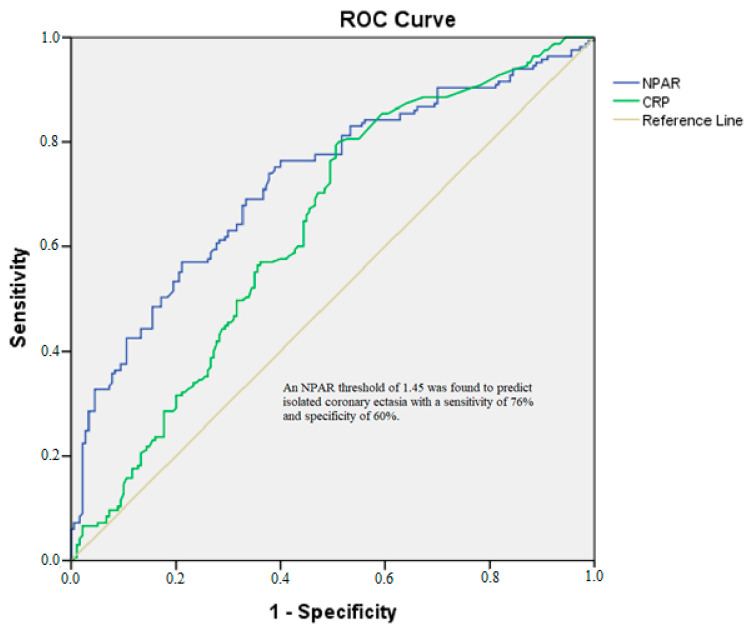
ROC analysis of NPAR for isolated CAE. CRP: AUC = 0.635, 95% CI (0.577–0.694). NPAR: AUC = 0.725, 95% CI (0.671–0.778).

**Figure 3 diagnostics-15-01638-f003:**
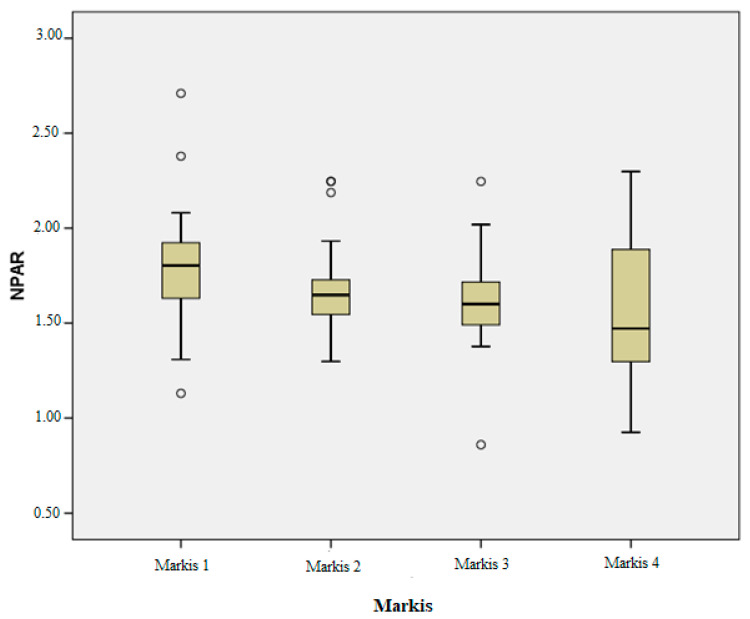
NPAR values of the Markis classes.

**Table 1 diagnostics-15-01638-t001:** Demographic distribution and laboratory findings of all patients.

	Overall (345)	CAE (165)	NCA (180)	*p*
	Mean ± Std. Deviation	Mean ± Std. Deviation	Mean ± Std. Deviation	
Age, years	59.99 ± 10.46	61.11 ± 10.16	58.97 ± 10.65	0.057
NPAR	1.28 ± 0.47	1.64 ± 0.31	1.42 ± 0.22	<0.001
Ejection fraction, %	61.13 ± 3.05	60.90 ± 3.07	61.50 ± 2.84	0.061
Plasma glucose	136.67 ± 54.20	141.77 ± 48.76	131.99 ± 58.49	0.094
Creatinine, mg/dL	0.81 ± 0.17	0.82 ± 0.16	0.80 ± 0.18	0.410
T cholesterol, mg/dL	197.97 ± 48.71	194.04 ± 44.26	201.56 ± 52.33	0.149
LDL-C, mg/dL	118.47 ± 39.33	115.65 ± 37.02	121.04 ± 41.28	0.204
HDL-C, mg/dL	43.64 ± 9.66	43.22 ± 8.89	44.03 ± 10.32	0.438
Triglyceride, mg/dL	177.82 ± 91.36	176.95 ± 117.08	178.62 ± 58.98	0.866
Albumin, g/dL	39.55 ± 3.08	39.08 ± 3.55	39.98 ± 2.52	0.008
hs-CRP, mg/L	5.50 ± 2.78	6.02 ± 2.76	5.02 ± 2.73	0.001
WBC count, ×10/μL	8.04 ± 1.96	8.20 ± 1.95	7.88 ± 1.96	0.135
Neutrophil count, ×10/μL	4.84 ± 1.54	5.25 ± 1.67	4.47 ± 1.29	<0.001
Lymphocyte count, ×10/μL	2.35 ± 0.94	2.12 ± 0.82	2.55 ± 0.99	<0.001
Monocyte count, ×10/μL	0.68 ± 0.23	0.73 ± 0.34	0.63 ± 0.21	0.105
Platelet count, ×10/μL	271.67 ± 66.94	278.86 ± 67.19	265.08 ± 66.22	0.056
Hemoglobin, g/dL	13.97 ± 1.78	14.05 ± 1.77	13.89 ± 1.79	0.405
NLR	2.41 ± 1.46	2.93 ± 1.75	1.95 ± 0.90	<0.001
PLR	133.51 ± 66.85	152.41 ± 77.20	116.19 ± 49.97	<0.001
	*n* (%)	*n* (%)	*n* (%)	*p*
Male, *n* (%)	235 (68.1)	116 (70.3)	119 (66.1)	0.404
DM, *n* (%)	136 (39.4)	67 (40.6)	69 (38.3)	0.666
Hypertension, *n* (%)	197 (57.1)	97 (58.8)	100 (55.6)	0.545
Current smoker, *n* (%)	130 (37.7)	65 (39.4)	65 (36.1)	0.530

Abbreviations: NCA, normal coronary artery; CAE, coronary artery ectasia; DM, diabetes mellitus; LDL-C, low-density lipoprotein cholesterol; HDL-C, high-density lipoprotein cholesterol; T, total; WBC, white blood cell; NLR, neutrophil-to-lymphocyte ratio; PLR, platelet-to-lymphocyte ratio; NPAR, neutrophil-percent-to-albumin ratio.

**Table 2 diagnostics-15-01638-t002:** Multiple logistic regression analysis showing independent predictors of isolated coronary artery ectasia.

	B	Sig.	95% Confidence Interval	
			Lower	Upper
NPAR	6.829	0.016	1.422	32.803
hsCRP	1.129	0.007	1.034	1.231
Neutrophil count	1.131	0.454	0.819	1.563
Lymphocyte count	1.012	0.961	0.624	1.642
NLR	1.046	0.859	0.638	1.714
PLR	1.005	0.096	0.999	1.012

Abbreviations: NLR, neutrophil-to-lymphocyte ratio; PLR, platelet-to-lymphocyte ratio; NPAR, neutrophil-percent-to-albumin ratio; hsCRP, highly sensitive C-reactive protein.

**Table 3 diagnostics-15-01638-t003:** Comparison of inflammatory biomarkers among Markis types I, II, III, and IV.

	Markıs 1(39)	Markıs 2(30)	Markıs 3(36)	Markıs 4(60)	*p* (Between Groups)
	Mean ± Std. Deviation	Mean ± Std. Deviation	Mean ± Std. Deviation	Mean ± Std. Deviation	
NPAR	1.78 ± 0.29	1.68 ± 0.24	1.62 ± 0.23	1.54 ± 0.35	0.001
Albumin, g/dL	38.64 ± 3.51	39.40 ± 3.93	39.28 ± 2.70	39.08 ± 3.87	0.816
hsCRP	5.44 ± 2.32	6.19 ± 2.64	6.40 ± 3.62	6.09 ± 2.48	0.471
Neutrophil count	5.77 ± 1.58	5.31 ± 1.50	5.63 ± 1.79	4.66 ± 1.59	0.004
Lymphocyte count	1.78 ± 0.73	1.94 ± 0.60	2.20 ± 0.86	2.38 ± 0.87	0.002
NLR	3.79 ± 1.94	3.01 ± 1.67	2.94 ± 1.69	2.31 ± 1.48	0.001
PLR	189.45 ± 104.56	172.05 ± 82.37	143.13 ± 59.69	124.07 ± 45.85	0.000

Abbreviations: NLR, neutrophil-to-lymphocyte ratio; PLR, platelet-to-lymphocyte ratio; NPAR, neutrophil-percent-to-albumin ratio; hsCRP, highly sensitive C-reactive protein.

## Data Availability

The original contributions presented in this study are included in the article. Further inquiries can be directed to the corresponding author.

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
