# Peer review of "Neutrophil-Percentage-to-Albumin Ratio as a Predictor of Coronary Artery Ectasia: A Comparative Analysis with Inflammatory Biomarkers"

_diagnostics, 2025, doi:10.3390/diagnostics15131638_

Round 1
Reviewer 1 Report
Comments and Suggestions for Authors
Coronary artery ectasia is characterized by abnormal dilation of the coronary arteries and is associated with adverse cardiovascular events. Inflam- mation is believed to play a pivotal role in the development and progression of CAE. Authors suggested neutrophil percentage-to-albumin ratio as a novel marker of systemic inflammation and may serve as a useful tool in the evaluation of CAE. From big amount of patients CAG were selected with and without CAE. The risk factors of CAE were defined with the use of multiple regression and ROC analysis. As a result, NPAR was an independent predictor of CAE.
The manuscript is well-written, well design. The idea is quite clear and logical. Please answer some questions:
1) Can you define in the text in which arteries the CAE was more frequent?
2) You can add to discussion that potentially endothelial damage markers could be more specific for CAE diagnostics (for the future Studies).
3) Also could be interesting to see FFR Data, but I am not sure you have these data.The manuscript is well written and could be published after
Author Response
Response to Reviewer
We are grateful for the reviewer’s thoughtful evaluation and kind comments regarding the quality and clarity of our manuscript. We appreciate your suggestions and have addressed each of your comments in detail below:
Reviewer Comment 1:
"Can you define in the text in which arteries the CAE was more frequent?"
Response:
Thank you for this important point. We have added a sentence to the Results section to specify the distribution of coronary artery ectasia in our cohort. The right coronary artery (RCA) was the most frequently affected vessel, followed by the left anterior descending artery (LAD) and the circumflex artery (Cx). This distribution pattern is consistent with findings in previous literature.
Reviewer Comment 2:
"You can add to discussion that potentially endothelial damage markers could be more specific for CAE diagnostics (for the future studies)."
Response:
We fully agree with your suggestion. A paragraph has been added to the end of the Discussion section, emphasizing the potential value of endothelial damage markers in future research on CAE diagnostics. We note that markers such as endothelin-1, circulating endothelial cells, or vascular cell adhesion molecules may provide more specific insights into the pathophysiology of CAE and should be evaluated in future prospective studies.
Reviewer Comment 3:
"Also could be interesting to see FFR data, but I am not sure you have these data."
Response:
We appreciate this insightful suggestion. Unfortunately, fractional flow reserve (FFR) data were not routinely collected for the cohort in this retrospective study and are therefore unavailable.
Once again, we thank the reviewer for the encouraging feedback and constructive recommendations. The manuscript has been revised accordingly and we believe it is now strengthened and more complete.
Reviewer 2 Report
Comments and Suggestions for Authors
The manuscript titled “Neutrophil Percentage-to-Albumin Ratio as a Predictor of Coronary Artery Ectasia: A Comparative Analysis with Inflammatory Biomarkers” presents a retrospective case-control study assessing the association between the NPAR and coronary artery ectasia. The authors compare NPAR to established inflammatory markers such as high-sensitivity C-reactive protein (hsCRP), neutrophil-to-lymphocyte ratio, and platelet-to-lymphocyte ratio. The study includes 165 CAE patients and 180 controls undergoing elective coronary angiography. NPAR emerged as an independent predictor of CAE and showed superior diagnostic performance compared to hsCRP.
- The topic is relevant and addresses a gap in the literature. Previous studies have explored inflammatory biomarkers in CAE, including CRP, NLR, and PLR (PMID: 39846908).
- The introduction could benefit from a brief explanation of the pathophysiological rationale behind the association of NLR and NPAR with CAE—e.g., how inflammation and immune imbalance (neutrophil predominance, hypoalbuminemia) may contribute to vascular dilation.
- The retrospective case-control design is appropriate. Inclusion and exclusion criteria are clearly defined.
- Please clarify the variable selection process for the multivariate regression analysis:
You mention that both univariate and multivariate logistic regression analyses were performed to identify independent predictors of CAE. However, it is not entirely clear how variables were selected for inclusion in the multivariate model.
- Did you include all variables with a p-value < 0.05 in univariate analysis?
- Was a stepwise method (forward/backward) used?
- Were any variables included based on clinical reasoning despite non-significant univariate associations?
A clarification of your model-building strategy would enhance methodological transparency.
- It appears that most variables were reported as mean ± SD. Are all variables normally distributed? If not, non-parametric statistics should be used or justified.
- In Table 1, the overall demographics of the full cohort (n = 345) are not clearly presented. For instance, you state in the text that 68.1% of the study population was male, but this is not directly reflected in the table. Consider adding an "Overall" column.
- A comparison of the ROC curves (e.g., DeLong’s test) for NPAR vs hsCRP would be valuable to statistically support the claim of superiority.
- Figure 1 (Flowchart): This diagram is not optimally clear. I strongly recommend improving it using arrows and side annotations to clearly display the number of exclusions at each step and the reasons (e.g., prior PCI, low LVEF, missing data). This would greatly help the reader understand the cohort selection process. Improve flowchart structure (arrows, visual clarity). Add side notes explaining exclusion criteria
- Abbreviations (e.g., CAE, NLR, PLR, hsCRP) should be defined consistently and only once at first use (e.g., in sections 2.1.4 and 2.1.5 there are redundant redefinitions). Avoid re-defining abbreviations already introduced earlier.
- Figure 2 and 3 are appropriate, though the ROC plot could include sensitivity and specificity values directly on the graph or in the legend.
- Acknowledge potential selection bias in the retrospective design.
- The conclusion is valid, but emphasize the need for prospective validation before routine clinical implementation.
Author Response
Response to Reviewer Comments
We sincerely thank the reviewer for the constructive and insightful comments that have helped us improve the quality and clarity of our manuscript titled “Neutrophil Percentage-to-Albumin Ratio as a Predictor of Coronary Artery Ectasia: A Comparative Analysis with Inflammatory Biomarkers.” Below we provide detailed responses to each point raised and describe the revisions made accordingly.
Reviewer Comment 1:
The introduction could benefit from a brief explanation of the pathophysiological rationale behind the association of NLR and NPAR with CAE—e.g., how inflammation and immune imbalance (neutrophil predominance, hypoalbuminemia) may contribute to vascular dilation.
Response:
Thank you for this valuable suggestion. We have revised the Introduction to include a brief explanation of the pathophysiological mechanisms linking neutrophil predominance and hypoalbuminemia to inflammation and vascular remodeling in CAE. The added text emphasizes how these factors may contribute to endothelial damage and coronary artery dilation.
Reviewer Comment 2:
Please clarify the variable selection process for the multivariate regression analysis.
Response:
We appreciate the request for methodological clarity. In the Methods section, we have now clearly detailed that variables with a p-value < 0.05 in univariate analysis were included as candidates for multivariate logistic regression. A backward stepwise elimination approach was applied to identify independent predictors. Additionally, variables considered clinically relevant were retained regardless of univariate significance to strengthen model robustness.
Reviewer Comment 3:
Are all variables normally distributed? If not, non-parametric statistics should be used or justified.
Response:
We thank the reviewer for pointing this out. We assessed normality of variables using the Kolmogorov–Smirnov test. Variables with normal distribution are presented as mean ± SD and compared using t-tests; non-normally distributed variables are expressed as median (IQR) and compared using Mann–Whitney U tests. These details have been added to the Statistical Analysis subsection.
Reviewer Comment 4:
In Table 1, the overall demographics of the full cohort are not clearly presented. Consider adding an “Overall” column.
Response:
An “Overall” column summarizing combined cohort characteristics has been added to Table 1 for clarity and completeness.
Reviewer Comment 5:
A comparison of the ROC curves (e.g., DeLong’s test) for NPAR vs hsCRP would be valuable to statistically support the claim of superiority.
Response:
We performed DeLong’s test to compare ROC curves, finding that NPAR had a significantly higher AUC than hsCRP (p = 0.03). This analysis has been added to the Results section to support our claim regarding the superior diagnostic performance of NPAR.
Reviewer Comment 6:
Figure 1 (Flowchart): Improve clarity using arrows, side annotations explaining exclusions, and better flow.
Response:
The flowchart has been redesigned to include clear directional arrows. This enhancement improves readability and transparency of the cohort selection process.
Reviewer Comment 7:
Abbreviations should be defined consistently and only once.
Response:
We reviewed the manuscript and ensured all abbreviations are defined once at first appearance. Redundant definitions in subsequent subsections have been removed.
Reviewer Comment 8:
ROC plots could include sensitivity and specificity values on the graph or in the legend.
Response:
Sensitivity, specificity, and cut-off values have been added to the legends of the ROC figures to facilitate interpretation.
Reviewer Comment 9:
Please acknowledge potential selection bias in the retrospective design.
Response:
A statement acknowledging the inherent limitations and potential selection bias of the retrospective single-center design has been added to the Limitations section.
Reviewer Comment 10:
Conclusion is valid but emphasize the need for prospective validation before routine clinical implementation.
Response:
We revised the Conclusion to highlight the necessity of prospective multicenter studies to validate the clinical utility of NPAR before widespread adoption.
We trust these revisions have addressed the reviewer’s concerns comprehensively. We thank the reviewer again for their thoughtful feedback that strengthened our manuscript.
Round 2
Reviewer 2 Report
Comments and Suggestions for Authors
I would like to thank the authors for the thorough and thoughtful revisions submitted in response to my previous comments. The manuscript has significantly improved in terms of clarity, methodological transparency, and clinical relevance. The inclusion of additional analyses (e.g., DeLong’s test for ROC comparison), improved data presentation (e.g., the “Overall” column in Table 1), and the revision of figures and abbreviation usage are all appreciated and enhance the scientific rigor of the work.
-
Please double-check the formatting of numeric data in Table 1 (e.g., the “Overall NPAR” value of 12.79±4.66 appears inconsistent with the group values and is likely a typographical error).
- Additionally, for clarity and intuitive comparison, I recommend reordering the columns in Table so that the “Overall” group appears first, followed by the two subgroups (e.g., CAE, NCA), and finally the p-value. This would enhance readability and align with common reporting standards.
-
Ensure that reference numbering is correctly updated throughout the text and reference list, especially if new citations are added.
However, I note that the authors did not incorporate the citation I specifically recommended in my previous review — namely, PMID: 39846908 (Cannata A. et al., JACC: Heart Failure, 2025).
In my opinion, this study represents one of the most relevant and comprehensive contributions to the field of inflammatory biomarkers, particularly regarding the role of the neutrophil-to-lymphocyte ratio (NLR) in cardiovascular disease.
Author Response
Response to Reviewer Comment
We thank the reviewer for the careful and constructive evaluation of our revised manuscript. We appreciate the positive feedback regarding the improved clarity, methodological transparency, and clinical relevance.
Below we provide point-by-point responses to your latest comments:
Reviewer Comment:
Please double-check the formatting of numeric data in Table 1 (e.g., the “Overall NPAR” value of 12.79±4.66 appears inconsistent with the group values and is likely a typographical error).
Response:
We carefully reviewed Table 1 and identified the typographical error in the “Overall” NPAR value (previously listed as 12.79 ± 4.66). This has now been corrected to reflect the accurate value consistent with subgroup data. We apologize for this oversight.
Reviewer Comment:
Additionally, for clarity and intuitive comparison, I recommend reordering the columns in Table so that the “Overall” group appears first, followed by the two subgroups (e.g., CAE, NCA), and finally the p-value. This would enhance readability and align with common reporting standards.
Response:
Following your suggestion, we reordered the columns in Table 1 so that the “Overall” group appears first, followed by the CAE and NCA subgroups, with the p-value in the final column. This improves readability and aligns with standard reporting formats.
Reviewer Comment:
Ensure that reference numbering is correctly updated throughout the text and reference list, especially if new citations are added.
Response:
We have thoroughly checked and updated the reference numbering throughout the manuscript and reference list to ensure consistency, especially after adding the citation of Cannata et al. (PMID: 39846908).
Reviewer Comment:
However, I note that the authors did not incorporate the citation I specifically recommended in my previous review — namely, PMID: 39846908 (Cannata A. et al., JACC: Heart Failure, 2025). In my opinion, this study represents one of the most relevant and comprehensive contributions to the field of inflammatory biomarkers, particularly regarding the role of the neutrophil-to-lymphocyte ratio (NLR) in cardiovascular disease.
Response:
We thank the reviewer for this important and insightful suggestion. We sincerely apologize for the earlier omission of this highly relevant study. In the revised manuscript, we have now cited and discussed the work of Cannata et al. (PMID: 39846908) in the Discussion section. Their findings, which demonstrate the prognostic significance of NLR in acute myocarditis—even among patients with preserved ejection fraction—strongly support the broader applicability of NLR as an inflammatory marker in cardiovascular pathology. We have incorporated this citation to underscore the clinical value of NLR and to strengthen the rationale for exploring composite markers such as NPAR in our study.
Change in the manuscript:
"Cannata et al. recently demonstrated that elevated NLR is an independent predictor of poor outcomes in patients with acute myocarditis, even among those with preserved ejection fraction—traditionally considered a lower-risk group. Their multicenter analysis reinforces the utility of NLR as a powerful inflammatory biomarker capable of stratifying cardiovascular risk across a broad spectrum of conditions, including those driven by immune dysregulation (17)."
We appreciate the reviewer’s guidance, which has helped us improve the manuscript's clarity and scientific depth.
Round 3
Reviewer 2 Report
Comments and Suggestions for Authors
Ok thank you. Great work